

# Making simulation results reproducible—Survey, guidelines, and examples based on Gradle and Docker

Wilfried Elmenreich, Philipp Moll, Sebastian Theuermann and Mathias Lux

Universität Klagenfurt, Klagenfurt, Austria

## ABSTRACT

This article addresses two research questions related to reproducibility within the context of research related to computer science. First, a survey on reproducibility addressed to researchers in the academic and private sectors is described and evaluated. The survey indicates a strong need for open and easily accessible results, in particular, reproducing an experiment should not require too much effort. The results of the survey are then used to formulate guidelines for making research results reproducible. In addition, this article explores four approaches based on software tools that could bring forward reproducibility in research results. After a general analysis of tools, three examples are further investigated based on actual research projects which are used to evaluate previously introduced tools. Results indicate that the evaluated tools contribute well to making simulation results reproducible but due to conflicting requirements, none of the presented solutions fulfills all intended goals perfectly.

## INTRODUCTION

Reproducibility of experimental results is fundamental in all scientific disciplines. Reproducing results of published experiments, however, is often a cumbersome and unrewarding task. *Casadevall & Fang (2010)* report that some fields, for example biology, are concerned with complex and chaotic systems which are difficult to reproduce.
At the same time, we would expect digital software-based experiments to be easily reproducible, because digital data can be easily provided and computer algorithms on these data are typically well-described and deterministic. However, this is often not the case due to a lack of disclosure of relevant software and data that would be necessary to reproduce results. Ongoing open science initiatives aim to have researchers provide access to data and software together with their publications in order to allow reviewers to make well-informed decisions and to provide other researchers with the information and necessary means to build upon and extend original research (*Ram, 2013*).

This article addresses two research questions (RQ) related to reproducibility:

**RQ1** *"To what extent is reproducibility of results based on software artifacts important in the field of computer science and in related research areas?"*
**RQ2** *"What tools can be used to support reproducibility?"*

Corresponding author
Wilfried Elmenreich,
wilfried.elmenreich@aau.at

*RQ1* addresses the aspects of the relevance of reproducibility to a researcher's field, willingness to contribute to making one's own work reproducible, and possible concerns. An online survey was designed to assess the current practice, subject awareness, and possible concerns. The focus of the survey was on the disciplines computer science, computer engineering, and electrical engineering and it addressed researchers at different positions in universities, research institutions, and companies. To answer *RQ2*, we present three examples where three different types of software projects are packaged to provide an accurate and easy possibility for reproducing results in a controlled environment and analyze how these solutions address the requirements derived from the survey.

The responses to our online survey confirm our initial assumption that the reproducibility of research results is an important concern in computer science research. One of the researchers' main reasons for publishing software artifacts along with scientific publications is improved credibility and reliability of results. Based on the survey's results presented in the section "Survey", we infer requirements and general guidelines assisting researchers in making their research reproducible in the section "Requirements and General Guidelines". Finally, we discuss how different tools comply with the created requirements and guidelines. We find that due to conflicting requirements, none of the presented solutions fulfills all intended goals perfectly. One of the most pressing challenges is to achieve long term availability of results while respecting licensing issues of required third-party dependencies. An in-depth discussion of open issues is elaborated in the section "One Tool to Reproduce them All?" and we conclude the article and highlight our major findings in the section "Conclusion".

## RELATED WORK

*Walters (2013)* notes that *it is often difficult to reproduce the work described in molecular modeling and chemoinformatics papers*. For the most part this is due to the absence of a disclosure requirement in many scientific publication venues thus far. *Morin et al. (2012)* report that in 2010 only three of the 20 most cited journals had editorial policies requiring availability of source code after publication. Fortunately, this situation is changing for the better, for example *Science* introduced a policy requiring authors to make data and code related to their publication available whenever possible (*Witten & Tibshirani, 2013*; *Peng, 2011*; *Hanson, Sugden & Alberts, 2011*). Commenting on this policy, *Shrader-Frechette & Oreskes (2011)* brought up the issue that although privately funded science may be of high quality, it is not subject to the same requirements for transparency as publicly funded science. Another obstacle is the use of closed-source tools and undisclosed software results in publicly funded research software development projects as discussed by *Morin et al. (2012)*. *Vitek & Kalibera (2011)* address the topic of repeatability and reproducibility for systems research and identify a particular difficulty for embedded systems due to companies being reluctant to release code and that implementations are often bound to specific hardware.

Focusing on the current state of reproducibility, ACM SIGCOMM Computer Communication Review (CCR) conducted a survey on reproducibility with 77 responses from authors of papers published in CCR and the SIGCOMM sponsored conferences

(*Bonaventure, 2017*). The responses showed that there is a good awareness of the need for reproducibility and a majority of authors either considered their paper self-contained or have released the software used to perform experiments. However, there were only few releases of experimental data or of modifications of existing open source software. The open question part of the survey indicated a need for encouragement for publishing reproducible work or for papers that attempt to reproduce or refute earlier results.

*Flittner et al. (2018)* conducted an analysis of papers from four different ACM conferences held in 2017. This study found that the type of artifacts can differ significantly between different communities. The analysis further indicates that even if researchers state that their work is reproducible, the majority of analyzed papers do not provide the complete toolset to reproduce all results. Most importantly, the study shows that published artifacts are indeed reused, which is why the authors encourage others to release artifacts.

A critical aspect when releasing artifacts is to decide on tools supporting researchers in the process of making research reproducible. Several papers report on case studies for data repositories in the context of reproducibility including fields such as geographic information systems (*Steiniger & Hunter, 2013*), astrophysics (*Allen & Schmidt, 2015*), microbiome census (*McMurdie & Holmes, 2013*), and neuroimaging (*Poline et al., 2012*). These examples are promising, but it cannot be expected that the approaches are going to be used beyond the field they have been introduced. Simflowny (*Arbona et al., 2013*) is a platform for formalizing and managing the main elements of a simulation flow, tied not to a field, but to a specified simulation architecture. The Whole Tale approach (*Brinckman et al., 2018*) aims at linking data, code, and digital scholarly objects to publications and integrating all parts of the research story. Other works focus on code and data management, such as *Ram (2013)* suggesting very general version control systems such as Git for transparent data management in order to enable reproducibility of scientific work. The CARE approach (*Janin, Vincent & Duraffort, 2014*) extends the archiving concept with an execution system for Linux systems, which also takes software installation and dependencies into account. Docker (*Boettiger, 2015*), which will be examined more closely in this article, provides an even more generic approach by utilizing virtualization for providing cross-platform portability. A tutorial for using Docker to improve reproducibility in software and web engineering research was published in *Cito, Ferme & Gall (2016)*. Reprozip by *Chirigati, Shasha & Freire (2013)* provided a packing and unpacking mechanism for Linux systems allowing the creation of a package from a computer experiment which can be unpacked on another target machine, including support for unpacking into a Docker image. In contrast to the work presented above, our work focuses on the researchers' requirements regarding reproducibility independent of the capabilities of individual tools. Based on survey responses, we infer requirements and guidelines for making research reproducible and further analyze how different tools for packaging software artifacts comply with the researchers' needs. We further identify limitations of current tools and raise awareness of researchers on the pros and cons of using different types of applications for making research reproducible.

## SURVEY

In computer science, a large amount of research is backed up by prototypes, implementations of algorithms, benchmarking and evaluation tools, and data generated in the course of research. A critical factor for cutting edge research is to be able to build upon the results of other researchers or groups by extending their ideas, applying them to new domains or by reflecting them from a new angle. This is easily done with scientific publications, which are mostly available online. While the hypotheses, findings, models, processes, and equations are published, the data generated and the tools used for generating data and evaluating new approaches are sometimes only pointed out but have to be found elsewhere.

Our hypothesis in that direction is that there is a gap between scientific publishing on one hand and the publication of software artifacts and data for making results reproducible for other researchers on the other. In that sense, we created a survey asking researchers in the computer science field for their approach and opinion.

### Methodology

The survey design is driven by our research question *RQ1* ("*To what extent is reproducibility of results based on software artifacts important in the field of computer science and in related research areas?*"). The survey consists of five parts. First, basic demographic information, including the type of research, the area of research, the typical research contribution, and the type of organization the researchers are working for, is collected. Second, the common practice of the researchers for publishing software artifacts and data is surveyed. Third, we focus on the researchers' expectations regarding the procedure of reproducing scientific results. Fourth, we ask for opinions on integrating the question of reproducibility in the peer review process. Finally, we collect additional thoughts with open questions.

Five-point Likert scales are used to indicate the level of agreement in the survey. For questions where Likert scales are not applicable, single-choice or multiple-choice questions (e.g., "*What are the typical results of your research work?*"), or numerical inputs without predefined range or categories (e.g., "*How much time (in hours) are you willing to invest to make the results of a paper reproducible?*") are used. Generally, we did not offer a "I don't know" or similar option. For single-choice and multiple-choice questions we discussed the nominal scales based on related work as well as the authors' experience. Pilots with people neither involved with the questionnaire nor taking part in the final survey were conducted to reduce the chance of leaving out important options. For the sake of completeness, custom values are allowed in addition to the given options, to allow researchers to report on their practice. Open-ended questions are only used where other types of questions might limit the spectrum of answers.

The survey was set up as an anonymous online survey, with no partial answers allowed as all questions were mandatory and only the final submission at the end of the survey would save the results. The survey was distributed via a scientific mailing lists and via personal contacts with the request to distribute the survey among colleagues[1]. The full survey and all responses are included in the Supplemental Material.

[1] The online survey was distributed on the following channels: Information-Centric Networking research group discussion list (https://www.irtf.org/mailman/listinfo/icnrg); the Google Group *comp. simulation* (https://groups.google.com/forum/#!forum/comp.simulation); the authors' Facebook and Twitter profiles; and via personal contacts.

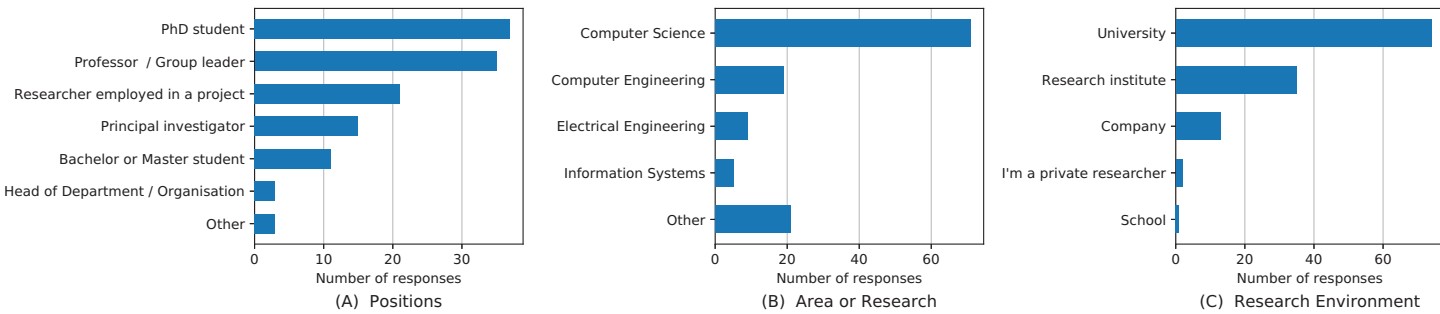

**Figure 1** Demographics of the survey participants including positions (A), area of research (B) and research environment (C). For presentation reasons, categories with less than three researchers were summarized as "Other" in (A) and (B).

## Demographics

In total, we received 125 responses, mostly from academic researchers. The demographics of survey participants are visualized in Fig. 1. Seventy-four out of the 125 participants were working or studying at a university and 35 of 125 of research institutes. Thirteen participants noted that they were mainly working for a company, two were private researchers, one from a school. With their position, 30% of the participants were PhD students, 28% were professors or group leaders, 17% worked as researchers within a project, 12% were principal investigators, and 9% were bachelor or master students at the time of the study. Three participants were heads of departments or organizations, and two participants indicated that they were postdoctoral researchers. Computer science or computer engineering was the area of research for 72% of the participants. Regarding the area of research, 7.2% of the participants came from electrical engineering, 4% from information systems, 3.3% from (applied) mathematics, and 1.6% from simulation. Furthermore, singular mentions were applied informatics, ciencias sociales, computational biology, computational biology/numerical simulations, computer networks, data analysis, economics, management, materials science, mathematical modeling, medical informatics, physics, scientific computing, and user experience. The population also includes researchers for whom publishing software is common practice; 28% of the participants have indicated that they have not published any software artifact at the time of the study.

## Survey result summary

Four aspects of the survey responses are analyzed. First, the relevance of reproducibility for the research community is analyzed. Second, we investigate what people are willing to do in order to achieve reproducible research. Third, we discuss the researchers' opinions on reproducibility in the peer review process. Finally, we highlight concerns regarding sharing scientific software artifacts.

Figure 2 summarizes the responses to questions showing the relevance of reproducibility in research. As can be seen, the majority of people wants to reproduce results from other researchers or groups: 103 of 125 indicated agreement. Even more (110 out of 125) considered reproducible results as added value for research papers. It can

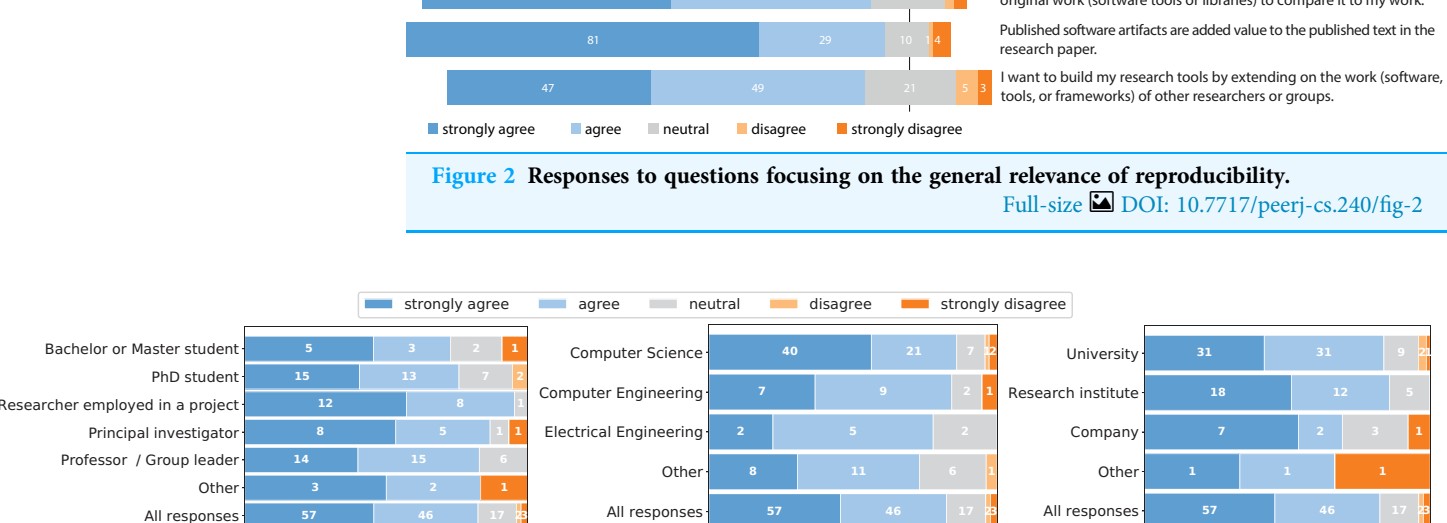

**Figure 2  Responses to questions focusing on the general relevance of reproducibility.**

**Figure 3  Responses to the question "*I want to reproduce the results of other researchers or groups from their original work (software tools or libraries) to compare it to my work.*" grouped by researchers' positions (A), research area (B) and research environment (C).**

be seen that the majority of researchers (96 out of 125) wants to build their research on the work of others, which requires others to share scientific artifacts.

It can be seen from the researchers' demographics in Fig. 3 that the relevance of reproducibility is independent of a researcher's position, research area, and research environment. The results of the question "*I want to reproduce the results of other researchers or groups from their original work (software tools or libraries) to compare it to my work*" were grouped by position, research area, and research environment. These distributions look very similar for all questions from Fig. 2. A full collection of graphical illustrations of these distributions is included in the Supplemental Material of the paper.

An open-ended question asking why software artifacts should be published yielded diverse answers. The most frequent answers were improvements in credibility and reliability of results, building trust, and improving understanding of the results of others. Besides, answers included the benefit of a practical approach by fostering task-based research, increasing visibility for your research by making tools available and open communication to foster research in general.

After showing the researchers' interests in reproducibility, which are aligned with the results from other published surveys, we now evaluate what researchers are willing to do to make their results reproducible for others and how much effort they are willing to invest to reproduce the results of others. Focusing on Fig. 4, we see that about half of the researchers typically try to reproduce the results of others by running their tools (53 out of 125). This again shows the demand for publishing scientific software artifacts.

The average amount of time participants would invest in making software of others work to reproduce results was 23.12 hours, neglecting two outliers who would spend $10^5$ and

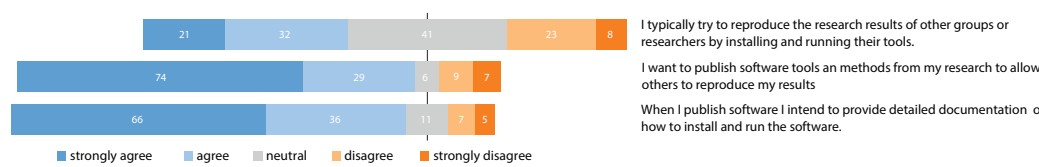

**Figure 4 Responses to questions focusing on what researchers are willing to do to achieve reproducible results or to share artifacts.**

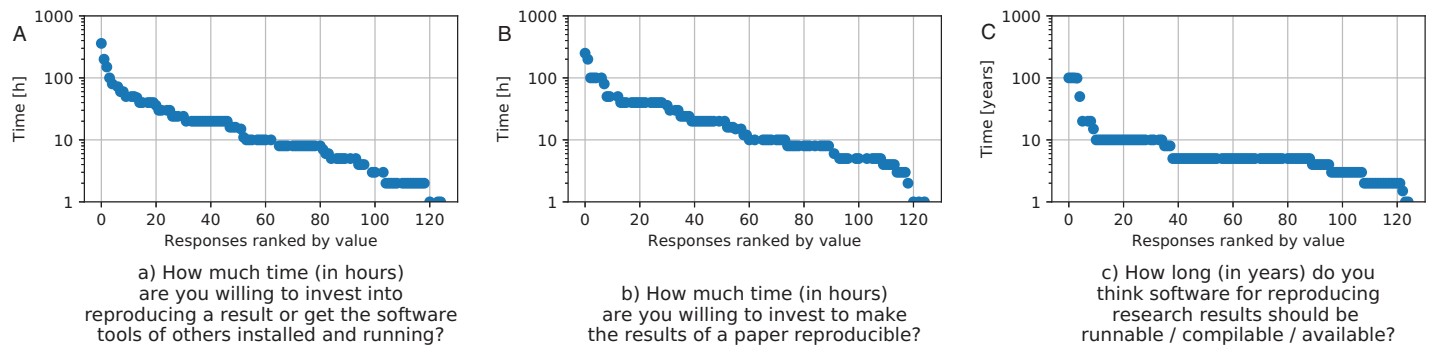

a) How much time (in hours) are you willing to invest into reproducing a result or get the software tools of others installed and running?

b) How much time (in hours) are you willing to invest to make the results of a paper reproducible?

c) How long (in years) do you think software for reproducing research results should be runnable / compilable / available?

**Figure 5 Responses to the above questions on how much hours researcher would invest into reproducing (A) and making reproducible (B) as well as how many years result should be reproducible (C).** Note that the *y*-axis is logarithmic.

---

[2] Using the range of mean ± 3 times standard deviation for outlier detection

$10^{35}$ or more hours[2]. The responses to the corresponding survey questions are visualized in Fig. 5A. Two participants noted that they would invest more than a month of work time (>160 h) to reproduce results of others, 10 participants noted that they would invest between a week and a month (41–160 h), 42 participants would invest up to a week, but less than a day (9–40 h), 47 would invest up to a day of work (1–8 h) and only three participants would not invest any time at all.

Most researchers agreed they would like to publish their software to aid reproducibility. The question of whether researchers want to publish their software tools to allow others to reproduce their results was answered with agreement from the majority of researchers (103 out of 125) with only 16 disagreeing. When publishing software, 102 out of 125 researchers intend to provide detailed documentation on how to install and run the published software artifact. The question of how many hours researchers want to invest in making their results reproducible led to an average of 24.4 hours. We excluded three outliers with answers of 1,000, $10^6$, and $10^{25}$ hours as we agreed that the answer of 1,000 hours—in other words 25 work weeks—and more is more likely to be a misunderstanding of the question and may include the original research work in addition to the extra work of making the results reproducible. The results can be seen in Fig. 5B. Summarizing the results in clusters results in two participants investing more than a month of work time (>160 h), seven participants would invest up to a month (41–160 h), 49 participants indicating they would invest up to week of work time (9–40 h), 38 participants reporting to invest up to a day (1–8 h), and only four indicating that they would not invest any time. Interpreting these numbers, we see that researchers are willing to invest more time to make their own research reproducible than to reproduce the results of others.

---

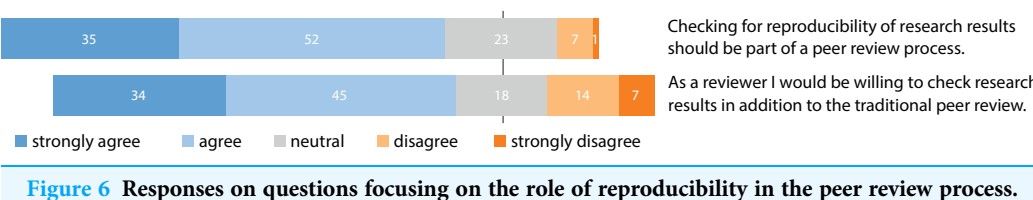

**Figure 6 Responses on questions focusing on the role of reproducibility in the peer review process.**

The results of a multiple-choice question asking for the typical composition of research results shows that software implementations and datasets are already common artifacts of today's research, indicating the potential utility of making research reproducible. Besides results in written form—107 researchers mentioned *published papers* and 37 participants *reports with detailed results*—a *software implementation* is part of the research results for 87 participants and 47 participants mentioned a *dataset* being part of their results.

Another important aspect for reproducible research is the long-term availability of results and artifacts. The effort of preparing and publishing software artifacts and results would ultimately be in vain if the artifacts later become inaccessible. Participants were asked about their opinion on how long results and necessary software artifacts should be available after initial publication. The results can be seen in Fig. 5C. With the exception of five outliers (with seven answers of 0 for not supporting reproducibility at all, as well as $10^6$, $10^9$, and $10^{25}$ years, which is too long a time for all currently known digital storage media to survive), the participants stated that software for reproducing results should be available for an average of 9.1 years. Summarizing through clusters 18 participants stated it should be from 0.5 to 2 years, 67 indicated it should be 3–5 years, 26 state 6–10 years, and nine think it should be more than 10 years available.

Asked about how they share research artifacts or make results reproducible, 90 out of 125 participants stated to have already published software at the time of their participation in the survey. Means of making their results reproducible were—multiple means could be specified—detailed instructions (68), make scripts (54), installation scripts (34), virtualization software (29), and container frameworks (15). There were two mentions of hosting web front ends as means of making results available and three mentions of public source code repositories as platforms for distribution.

Now that we are aware of current practices for making results reproducible, we focus on the role of reproducibility in the peer review process. Our assumption is that testing for reproducibility during the peer review process could enhance the credibility of published results and thereby increase the quality of a paper. This opinion is shared by the survey participants as visualized in Fig. 6: A total of 87 out of 125 participants stated that checking for reproducibility should be part of the peer review process. Furthermore, 79 out of 125 participants would be willing to check results in addition to the traditional peer review process.

Here, differences among different positions and research areas can be found (see Fig. 7). When focusing on the researchers' position, nine out of 10 bachelor or master students showed agreement, with none indicating disagreement. Principal investigators indicated the lowest agreement. Differences can also be seen regarding different research areas. Researchers from computer engineering showed the least agreement, whereas electrical

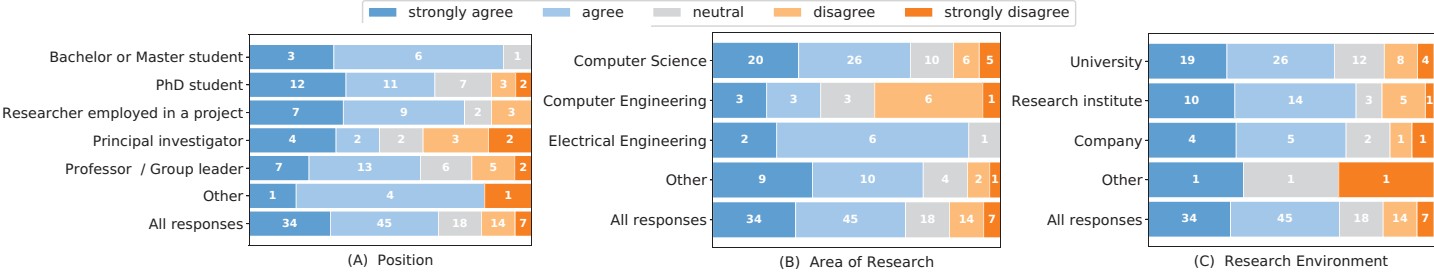

**Figure 7 Responses to the question "*As a reviewer I would be willing to check research results in addition to the traditional peer review.*"** grouped by the researchers' position (A), research area (B) and research environment (C). 

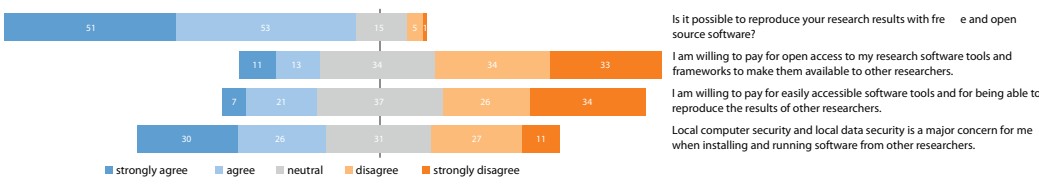

**Figure 8 Responses to questions focusing on additional concerns when publishing scientific artifacts.** 

engineers indicated the most agreement. Researchers from other research areas, including computer science, indicated a similar interest. No differences between different research environments were identified.

When analyzing the survey results on the researchers' concerns regarding publishing scientific software artifacts, we can see that the traditional payment models of scientific publishers used for research papers are seen as critical for publishing software artifacts. Figure 8 shows that 104 out of 125 researchers indicate that their results can be reproduced with free and open source software. This goes hand in hand with researchers' reluctance to pay for publishing or accessing software artifacts. Only 24 out of 125 researchers are willing to pay for making software tools, frameworks, and subsequently their results to be available to other researchers. A few more, but still only 28 out of 125 researchers indicate agreement with paying for being able to reproduce the results of others. These responses indicate the importance of possibilities for sharing software artifacts free of charge regardless of the platform. Moreover, even researchers willing to pay for software, might face problems due to closed-source components or other limitations.

Continuing in this vein, we asked why results cannot be reproduced using open source tools. A total of 50 participants indicated the use of paid-for programming language environments, 35 the use of licensed operating systems, 19 the use of copyrighted materials, and 11 the use of commercial tools.

Computer security, when installing programs from others, is not a major concern for 69 out of 125 participants, which is alarming when reflecting on possible security issues. An explanation could be that the researchers' awareness is low because they themselves would not harm others and believe others to be benevolent as well. However, this mindset does not account for security issues that do not originate from other researchers, but from

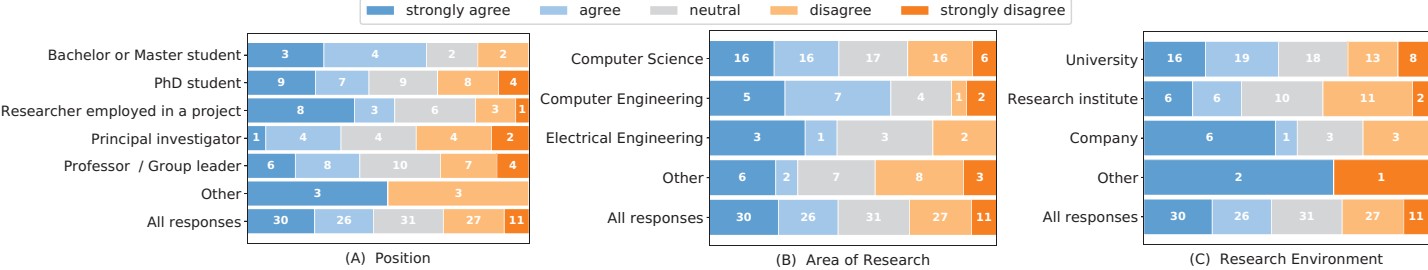

**Figure 9** Responses to the question "*Local computer security and local data security is a major concern for me when installing and running software from other researchers.*" grouped by the researchers' position (A), research area (B) and research environment (C).

used third-party libraries. Therefore, software from unknown sources, or with unknown dependencies, should always be handled with care.

We further see that security awareness depends on the researchers' position (see Fig. 9). Undergraduate and master students indicate the highest awareness of security risks, while professors and principal investigators the lowest. A possible interpretation is that researchers in higher positions neglect security issues because of the high pressure to progress research. Students, in contrast, focus on smaller tasks and complete them more carefully. Regarding security awareness across different research areas, computer engineers have the highest awareness with 12 out of 19 researchers indicating agreement on the question "*Local computer security and local data security is a major concern for me when installing and running software from other researchers.*" For other fields, the awareness or lack thereof is almost equally low.

Besides economical and security concerns, we also asked researchers about additional reservations. A multiple-choice question on major concerns showed that when installing and running software from other researchers the ease of installation is a prominent topic. This questions allowed for multiple choice as well as an other option, where participants could voice their concerns. Answers included:

- Ease of the installation (without major barriers) (104 mentions),
- Hardware requirements like computation power, memory, or specialized equipment (71 mentions),
- License issues (72 mentions),
- Size of the download and installation (27 mentions),
- Used harddisk space after installation (two mentions),
- I don't see additional concerns (eight mentions),
- Other (with the option of giving text here).

Five other answers were entered:

- "I am sure it does not run on the first try nine out of 10 times,"
- "External dependencies and their updateability/patchability in case of security fixes (should never depend on the initial publisher for third party libraries because they'd

have to maintain their old packages for a long time); Also important: Downards (sic!) compatibility of "new" versions with old data & tools,"

- "Analytical reproducability (sic!) & mathematical clarity (or correctness) is my main concern,"
- "Conflicting versions of additional required software,"
- "complex build dependencies."

A final open-ended question was about reservations toward publishing data and software: "*I have reservations for publishing software artifacts and data in research because ...*" For analysis following the approach of open coding the answers were labeled manually by the authors and the assigned labels were discussed until agreement was reached. The most common cluster of answers noted legal or privacy issues (14). Others pointed out the additional effort needed (8), commercial interests (8), missing reward or support for doing so (3), and that publishing artifacts is not part of the job, that is, not supported by the group or organization (2)[3].

Regarding the aforementioned legal issues, it would be an interesting hypothesis that researchers would be more willing to share if legal issues and efforts are reduced. This may be achieved by license constraints (only licenses others can build upon) or exceptions for publishing research (leaving license issues aside for research by general agreement).

## Correlation analysis

Given the Likert scales for the answers we did investigate the correlation (Spearman's rank correlation) between answers to see if (i) intuitive and expected correlations exist and (ii) new and surprising correlations can be found. Table 1 shows all correlations with $|\rho| > 0.4$[4]. The strongest correlation to be found with a coefficient of $\rho = 0.734$ and a $p$-value $< 0.0001$ was between the questions "*Checking for reproducibility of research results should be part of a peer review process*" and "*As a reviewer I would be willing to check research results in addition to the traditional peer review.*" Hence, people who stated to be willing to do reproducibility checks were more likely to find the idea of a review process with mandatory reproducibility checks attractive.

Another strong correlation ($\rho = 0.723$, $p < 0.0001$) was found between the questions "*How much time (in hours) are you willing to invest into reproducing a result or get the software tools of others installed and running?*" and "*How much time (in hours) are you willing to invest to make the results of a paper reproducible?*" With that correlation one can hypothesize that researchers with reproducibility in mind invest time in reproducing results as well as making their results reproducible.

A less strong but still rather interesting correlation ($\rho = 0.55$, $p < 0.0001$) was found between "*I am willing to pay for open access to my research software tools and frameworks to make them available to other researchers.*" and "*I am willing to pay for easily accessible software tools and for being able to reproduce results of other researchers.*" So with the overhead of participants not willing to pay for access and publishing of in context of reproducibility as indicated in Fig. 8, it is likely that researchers either like the idea of either paying for both, publishing and access, or none.

---

[3] The raw data with all the responses is included in the Supplemental Material.

[4] Correlations of $|\rho| < 0.4$ are generally considered as poor or weak correlations and hence not included in the table.

**Table 1** Correlations in the survey answers with |ρ| > 0.4 using Spearman's rank correlation.

| Questions | ρ | *p*-value |
|---|---|---|
| Checking for reproducibility of research results should be part of a peer review process. | 0.734 | <0.0001 |
| As a reviewer I would be willing to check research results in addition to the traditional peer review. | | |
| How much time (in hours) are you willing to invest into reproducing a result or get the software tools of others installed and running? | 0.723 | <0.0001 |
| How much time (in hours) are you willing to invest to make the results of a paper reproducible? | | |
| I am willing to pay for open access to my research software tools and frameworks to make them available to other researchers. | 0.550 | <0.0001 |
| I am willing to pay for easily accessible software tools and for being able to reproduce results of other researchers. | | |
| I want to publish software tools and methods from my research to allow others to reproduce my results. | 0.490 | <0.0001 |
| When I publish software I intend to provide detailed documentation on how to install and run the software. | | |
| I want to reproduce the results of other researchers or groups from their original work (software tools or libraries) to compare it to my work. | 0.482 | <0.0001 |
| I want to build my research tools by extending on the work (software, tools or frameworks) of other researchers or groups. | | |
| I want to reproduce the results of other researchers or groups from their original work (software tools or libraries) to compare it to my work. | 0.477 | <0.0001 |
| I typically try to reproduce the research results of other groups or researchers by installing and running their tools. | | |
| When I publish software I intend to provide detailed documentation on how to install and run the software. | 0.412 | <0.0001 |
| I want to build my research tools by extending on the work (software, tools or frameworks) of other researchers or groups. | | |
| When I publish software I intend to provide detailed documentation on how to install and run the software. | 0.410 | <0.0001 |
| I typically try to reproduce the research results of other groups or researchers by installing and running their tools. | | |
| Published software artifacts are added value to the published text in the research paper. | 0.408 | <0.0001 |
| I want to build my research tools by extending on the work (software, tools or frameworks) of other researchers or groups. | | |

## Threats to validity

While a minor bias is assumed to be caused by the study's title as participants may have been attracted by the title if they could identify with the topic of reproducibility, it is still valid to discuss the implications of the findings.

One possible limitation of the survey is the missing geographical distribution of the participants. We did not include questions on where participants are located or work primarily, and did not collect IP addresses. Hence, we cannot conclude if the survey result indicates a global trend, or if the preferences of researchers from different geographic regions differ. Similarly, a possible gender gap of the survey's participants can not be evaluated. For single and multiple choice questions with a pre-defined answer set in the survey, the set of answers can introduce a certain bias to the results. Therefore, it was decided to avoid such questions if the risk of bias was high. In that sense, we also avoided quantizing numeric input, for example, the hours people spend making their work reproducible. If it was necessary, we always included an open-ended answer option. The pilot, survey of related work, and critical reflection by the authors were used as tools to minimize the bias. In one single case, that is, the question "*Which of the following are*

*additional concerns when installing and running software from other researchers?*" the open-ended answer option showed that at least one pre-determined answer was missing. Several participants noted complex build dependencies (also mentioned as conflicting versions of additional required libraries or external dependencies) are likely to be another major concern.

Participants could have had overlaps in the categorization of positions, for example a person could be a PhD student and an employed researcher in a project at the same time. In this case participants might have selected the category randomly or selected the category they appreciate more. Despite this, as long as no intentional or unintentional mistakes are made in the answers, each category will contain samples that are member of this category. The survey also includes bachelor and master students, where it is not guaranteed that they are involved in research projects. However, due to the dissemination channels we used for advertising the survey, we assume that the participants are invested in research and reproducibility. This is supported by the study where nine out of 11 bachelor or master students indicate to have already published software artifacts.

English as the only language for the survey might be a further limitation. Nevertheless, English is the working language of the target audience, and consequently, we assume the influence by the survey language to be negligible.

## REQUIREMENTS AND GENERAL GUIDELINES

The survey results indicate that a majority of researchers of all levels consider reproducibility as very relevant. There is further a strong interest in doing work to make one's own results reproducible, a strong interest to use results of others for comparison to own work, and to some extent, a motivation to try to reproduce work for review purposes.

To achieve this, it is necessary to make all information that is necessary to reproduce the results available together with a publication. Additionally, the effort necessary to reproduce the results needs to match the value of doing the work. Work reproducing or refuting previous results is overall much less appreciated than original work, so the effort a researcher is willing to invest in order to reproduce previous results is much lower than the effort they are willing to put in to produce new work. On the other hand, when planning to build own research on top of other results the investment can be higher. The most critical case is in reviewing, when reproducibility is intended to be checked as part of the reviewing process. Reviewers have a strict timeline to perform their review, so there is a need for a straightforward, mostly automated process to reproduce results. Moreover, despite contributing to verifying the results of a paper, reviewers are not mentioned in connection with the work. As reviewers work voluntarily, they are probably the least motivated to reproduce results.

Moreover, in our study researchers have responded critically to commercial systems introducing payments, either from the publishing researcher side or from the consumer side. A majority of participants also name security as a concern in this context, which highlights an issue to be addressed for researchers being security-aware as well as for those who are less concerned about security.

In order to address these issues, the following guidelines are proposed:

- Code, data, and information on how to conduct an experiment should be gathered in a single place (a single container), which can be found in connection with the paper.
- The reproduction process should be highly automated (for example by an easy to handle build and execution script).
- To address security issues, the published artifacts should be provided as source code and scripts allowing for running the code in a virtual environment should be provided.
- Commercial libraries and other components that require reviewers to pay for access should be avoided.
- Since research papers tend to create some interest even long after they have been published, it is necessary to ensure that software and environment for the reproduction process remain available, either by packing all necessary components into a container or by referring to well-archived open source tools.
- The time and necessary information to reproduce results should be tested with an independent person. Unless the size of the project requires it, the reproduction process should take at most 2 days.

## EXISTING TOOLS

Most tools for sharing software artifacts are also used in the development of software artifacts. These could be either tools for simple tasks such as compiling software projects, but also more complex tools for tasks such as automated dependency installation and software packaging. To prevent unnecessarily complex configuration, it is wise to select tools based upon the complexity of the software artifact. Software artifacts which are complex to run require more sophisticated tools with high levels of abstraction, whereas simple artifacts do not require complex tools to run.

In this section, we tackle our second research question by presenting four open source tools for sharing software artifacts, ranging from tools for compiling simple artifacts to sophisticated frameworks for sharing self-contained software environments. The tools have been selected despite of their different scopes because of their potential to support reproducible research. It has to be noted that a complex project might even incorporate multiple tools, for example a build system within a virtual environment.

We begin with a discussion of simple tools, such as *CMake*, which are used for build management and continue by discussing tools utilizing a higher level of abstraction. For discussion purposes, well-known tools, each representing a class of tools with similar functionality, were selected. Discussed pros and cons are valid not only for the discussed tool itself, but for the complete class represented by the tool. Finally, we summarize the features of the different tools and discuss the importance of their benefits, according to the survey results presented in the section "Survey".

### CMake

CMake is a cross-platform build tool based on C++. It is designed to be used with native build environments such as *make*. Platform-independent build files are used to generate

compiler-specific build instructions, which define the actual build process. Main features of CMake are tools for testing, building, and packaging software, as well as the support of hierarchical structures, automatic resolution of dependencies and parallel builds.

One drawback of CMake and similar build management systems is that required libraries or other dependencies of software artifacts must be available and installed in the required version on the host system in order to successfully build the project. This could lead to extensive preparations for a build which is mandatory for executing software artifacts.

CMake has been chosen for discussion because it is one of the most used tools of this type. Tools with similar functionality are *configure scripts*, the *GNU Build System* and the *WAF* build automation system.

## Gradle

Gradle is a general purpose build tool based on Java and Groovy. Gradle integrates the build tools Maven as well as Ant and can replace or extend both systems. Main features of Gradle are the support for Maven repositories for dependency resolution and installation and the out of the box support for common tasks, that is, building, packaging and reporting. Gradle supports multiple programming languages, but has a strong focus on Java, especially as it is the recommended build system for Android development.
An integrated wrapper system allows to use Gradle for building software artifacts without installing Gradle. Dependency installations and versions are maintained automatically. If a build requires a new version of a library, it is downloaded and installed automatically.

The automated dependency installation is a great benefit of Gradle, although there are still some challenges to overcome. One issue is that automated dependency installation only works, if the required libraries are offered in an online repository. If the required dependency is removed from the online repository, building any software depending on this library becomes impossible.

For other programming languages, tools with similar functionality are available, that is, the *Node Package Manager* for JavaScript projects or *pip* for Python projects.

## Docker

The open source software *Docker* allows packaging software applications including code, system tools, and system libraries into a single *Docker image*. The resulting image can be published, downloaded and executed on various systems without operating system restrictions in a virtualized environment. This way, an application embedded in a Docker image will execute in a predefined way, independent of the software environment installed on the host computer. The only requirement for the host system is the installed Docker engine.

A Docker image is a kind of lightweight virtual machine image. It could contain the runtime environment for a single application with or without graphical user interface, but it could also contain a ready to deploy server application for web services or even environments for heavy calculations or simulations. When running the Docker image, a *Docker container* is launched. A Docker container can be seen as an isolated runtime

environment, which uses the kernel of the host operating system and thereby becomes more lightweight than traditional virtual machines. A running Docker container can be accessed via a terminal or a graphical user interface allowing for a broad range of applications.

Docker images can be shared in two different ways. The first way is to export a running container including all files and executables as image and to share it as a single file. This file can be large in size but is fast to launch by others. The second way is to create a so-called *Dockerfile*. Dockerfiles contain the building instructions for Docker images. These instructions include commands for installing required dependencies and for installing the shared software artifact itself. When building a Docker image from a Dockerfile, all instructions from the Dockerfile are automatically executed. This leads to a small Dockerfiles, but a more complex import process. In addition, when using Dockerfiles, all dependencies need to be available either in online repositories, or locally on the machine building the image. The commercial *Docker Hub* platform (https://hub.docker.com/, last visited 2019-10-10) streamlines the process for sharing Docker images. Docker provides tools to share images on Docker Hub and to download images from Docker Hub via the command-line. Docker Hub offers the possibility of sharing public Docker images without download restrictions for free, but also paid plans allowing creating private repositories for sharing images among small groups.

The major difference between Docker and the previously presented tools is that Docker is not usually used for the development of an artifact. In most cases, a Docker image is created for sharing a predefined environment in a team. This means that the image is created and the software artifact is deployed in the container afterward.

An alternative to Docker is using Linux containers, which allow to run multiple isolated Linux systems on a single host.

## VirtualBox

VirtualBox is an open source software for the virtualization of an entire operating system. VirtualBox emulates a predefined hardware environment, where multiple operation systems, like Windows, Mac OS, and most Unix Systems can be installed. The installed operating system is stored as persistent image, which allows the installation and configuration of software. Once the image is created, it can be shared and executed on multiple machines.

As mentioned before, VirtualBox emulates the entire hardware of a computer resulting in higher execution overhead as well as higher setup effort. Before the scientific software artifact can be installed in a VirtualBox container, an operating system and all dependencies have to be installed.

A non-open source alternative to VirtualBox is VMWare, which has similar functionality.

## Comparison of analyzed tools

After the presentation of selected tools in the last section, we now want to compare their features for sharing scientific software artifacts. As criteria for the comparison, we focus in this section on important aspects of software for researchers, according to the survey

**Table 2 Comparison of tools for sharing scientific software artifacts.**

| Tool | CMake | Gradle | Docker | VirtualBox |
|---|---|---|---|---|
| Security | No security mechanisms | No security mechanisms | Sandboxed environment | Sandboxed environment |
| Supported platforms | Linux, MacOS, Windows | Java VM | Linux, MacOS, Windows | Linux, MacOS, Windows |
| Required knowledge for sharing | Little | Little | Moderate | Little |
| Effort for sharing | Little | Little | Moderate | High |
| Required knowledge for installation and execution | Moderate | Moderate | Little | Little |
| Effort for installation and execution | Moderate/high | Little | Little | Little |
| Size of shared object | Small | Small | Up to multiple GBs | Up to multiple GBs |
| Limitations | Installation could be exhausting | Specific Gradle project structure recommended | GUI requires extra effort | Images always include the entire operating system |

presented in the section "Survey". Table 2 briefly summarizes our findings; a description of each criteria is found throughout this section. The ratings in Table 3 are based on qualitative comparisons, as well as on our experience from using the tools for making three different research projects reproducible, as elaborated in the section "Examples".

### Security

As indicated by the survey, local computer and data security is a major concern for many researchers. Some software artifacts require administrator access rights on the local machine in order to be executed. These access rights allow malicious behavior, which could lead to unwanted consequences on the local machine or on the local network.

VirtualBox and Docker execute software artifacts in sandboxed environments and therefore allow the secure execution of software artifacts. Tools like CMake and Gradle do not offer this security mechanism. When executing a shared software artifact from untrusted sources, a sandboxed environment is recommended.

### Supported platforms

CMake, Docker, and VirtualBox are compatible with most Linux platforms, recent versions of MacOS, and selected versions of Windows 10. Gradle works as long as the Java Virtual Machine is available. Besides this platform support it has to be kept in mind that the software artifacts itself could require a certain operating system. This problem can be mitigated through virtualization of Docker and VirtualBox.

### Required knowledge for sharing

If a build management tool is used in the development of a scientific software artifact, we assume that the researchers are familiarized with the build management tool during the development phase. Therefore, no additional knowledge for the researcher who is sharing the artifact is required. VirtualBox also does not require a lot of additional background information. Everybody who is able to install an operating system is able to share a software artifact embedded in a VirtualBox image. The terminology of Docker seems to be confusing at first glance, requiring some time to become familiar with Docker concepts.

### Effort for sharing

CMake, Gradle, and other build management systems are intended to define a standardized build process. If a build management system is used during the development of the scientific software artifact, no additional effort arises for sharing. The configuration file for the build management system can be shared along the source code of the software artifact.

Docker and VirtualBox are usually not directly involved in software development. In most cases, a Docker or VirtualBox image has to be created explicitly for sharing the software artifact. The structured process of building a Docker container allows easy reuse of existing Docker containers for other software artifacts. In the case of VirtualBox, the whole VirtualBox image has to be shared on a file server. Docker images can be shared on the free to use Docker Hub or on a file server. Alternatively, a Dockerfile, which contains the building instructions for a Docker image, can be created and shared as a text file. However, using a Dockerfile requires all dependencies being available in repositories, adding additional complexity to the overall process.

### Required knowledge for installation and execution

Researchers are often not familiar with the tools used for the creation of software artifacts. Reading the documentation of build management tools can be exhausting and time-consuming for the short test of an artifact. CMake and Gradle require some knowledge in order to build a software artifact, especially if errors appear.

VirtualBox and Docker are easier to use. If a Docker image is hosted on DockerHub, a single command is sufficient for downloading and running the image. If this command is provided, no additional knowledge is required. Due to a graphical user interface, running a VirtualBox image is even easier.

### Effort for installation and execution

According to the survey results, ease of installation is a major consideration for most researchers (104 of 125 participants). Regarding the installation of the used tool itself, Gradle has the lowest requirements. The Gradle Wrapper allows installing dependencies and the build of artifacts without installing Gradle itself. For installing and executing the shared software artifact, the highest effort arises when using CMake, where required dependencies have to be installed manually. For building and executing software artifacts with Gradle only a few commands are required. Docker and VirtualBox require the least effort; the shared image only needs to be executed.

### Size of shared object

When using CMake or Gradle, the source code of the software artifact and the configuration file of the build management tool have to be shared, which usually leads to small shared objects.

The shared image of Docker or VirtualBox has to contain the source code and all other tools which are required for execution, such as the operating system. This results in large shared objects, in some cases the size of a Docker image exceeds one Gigabyte.

Alternatively, Docker provides an option allowing for smaller shared objects—Dockerfiles. A Dockerfile contains only text instructions for building a Docker image. Therefore, the size of a Dockerfile is only a few kilobytes, but once executed, Docker automatically pulls the artifact's source code from provided repositories and builds the software artifact, resulting in a large Docker image on the local machine. Nevertheless, the size of the download is not a major concern for the majority of the survey participants.

### Limitations

All analyzed tools have limitations. CMake is a lightweight tool for software development, but the effort for installing the dependencies of a software artifact could be extensive. Furthermore, it is only applicable for a handful of programming languages such as C or C++.

When Gradle is chosen as build system early in the development phase, it is perfectly suited for Java projects. Using Gradle for existing projects can be cumbersome because it requires additional configuration for projects that do not comply with Gradle's default project structure. Especially for researchers that are not familiar with Gradle, the time spent for this additional configuration step should not be neglected.

Docker is perfectly suited for command-line or web applications, which is the case for a huge amount of scientific software artifacts. Additional configuration is required to support GUIs of desktop applications. FREVO (see the section "FREVO"), used in one of our examples, demonstrates GUI support for desktop applications with Docker.

VirtualBox is applicable for all types of software artifacts, but the overhead of creating and sharing a VirtualBox image could be huge. For sharing an artifact, independent of its size and complexity, a complete operating system has to be installed and shared.

## EXAMPLES

After introducing background information in the last sections, three examples are presented analyzing the applicability of various tools for sharing software artifacts. Three scientific artifacts from different computer science research areas allowed us to focus on various types of artifacts with different build systems and procedures for sharing. The first example—Stochastic Adaptive Forwarding—is a simulation scenario, which can be executed on a command line in order to conduct performance evaluations. Second, FREVO is a simulation tool, mainly controlled via a graphical user interface. The third example—LireSolr—is a server-based application used for image retrieval.

### Stochastic Adaptive Forwarding

Stochastic Adaptive Forwarding (SAF) (*Posch, Rainer & Hellwagner, 2017*) is a forwarding strategy for the novel Internet architecture Named data networking (NDN) (*Zhang et al., 2014*). Forwarding strategies in NDN are responsible for forwarding packets to neighboring nodes and therefore select the paths of traffic flows in the network.

The Network Forwarding Daemon (NFD) implements the networking functionalities of NDN. It is written in C++ and uses the WAF build automation system. The network simulator ns-3/ndnSIM (*Mastorakis et al., 2016*) is used for testing purposes, which also uses the WAF build system. For testing SAF in the simulation environment three steps are required: (i) Installation of the NFD; (ii) installation of the network simulator

ns-3/ndnSIM and finally (iii) patching SAF into a compatible version of the NFD. The installation of SAF was tested and analyzed in the standard way by using WAF and Docker.

### SAF with WAF

The standard way of developing NDN forwarding strategies is by using the WAF build automation system. The functionality of the WAF build system is similar to the functionality of CMake. This means that WAF automatically resolves dependencies, but the installation of dependencies must be performed manually. Although extensive installation instructions were published (https://github.com/danposch/SAF, last visited 2019-07-08), it is tricky to install the simulator and its dependencies. Furthermore, there are slightly undocumented differences when installing the NDN framework on different Unix versions. Once the NDN framework is compiled in the correct version, it is easy to patch SAF. Nevertheless, it can take up to several hours to initially install and compile the NDN framework with SAF.

### SAF with Docker

Named data networking and SAF are licensed under GPL V3, meaning that there are no legal concerns over packaging the software. Technically, Docker provides two options for creating and sharing images. The first is to check out a preconfigured image like Ubuntu Linux from the Docker website and connect to it via terminal. All required changes can be made in the terminal and finally persisted with a commit. The altered image can be shared via the Docker website or as binary file. The second possibility to create the image is by using Dockerfiles. These files contain simple creation instructions for images and can be shared easily due to their small size. To build an image, the Dockerfile can be executed on any host with the Docker framework installed. Both variants were tested for SAF. The resulting images, containing all dependencies and the compiled software artifacts, have a size of about 4.6 GB with the size of the Dockerfile being about two KB. Using the precompiled image (https://hub.docker.com/r/phmoll/safprebuild/, last visited 2019-07-08), running the image only takes an instant. The execution of the Dockerfile takes, depending on the Internet connection and the computing power of the host system, between 15 and 60 min. Once the image is running, the results of the paper can be reproduced or new experiments with SAF can be conducted using the provided network simulator.

## FREVO

FREVO (*Sobe, Fehérvári & Elmenreich, 2012*) is an open source framework to help engineers and scientists in evolutionary design or optimization tasks to create a desired swarm behavior. The major feature of FREVO is the component-wise decomposition and separation of the key building blocks for an optimization task. This structure enables the components to be designed separately allowing the user to easily swap and evaluate different configurations and methods or to connect an external simulation tool. FREVO is typically used for evolving swarm behavior for a given simulation (*Fehervari & Elmenreich, 2010*; *Monacchi, Zhevzhyk & Elmenreich, 2014*). FREVO is a mid-sized project with 50k lines of mostly Java code, having a graphical interface as well as a mode for pure command line operation, for example, to be used on a simulation server.

The component-based structure allows to easily extend and remove components (e.g., a simulation, a type of neural network, an optimization algorithm), which sometimes creates some effort in newly setting up FREVO.

FREVO was tested and analyzed with the following three tools:

### FREVO with build script

Until recently, FREVO was provided as a download zip file (http://frevo.sourceforge.net/, last visited: 2019-07-08) that included sources of the main program and additional components together with an ant build file. However, there had been problems in the past with different language versions of Java. A further problem can be dependencies on third party tools or libraries, which are not automatically maintained by this type of build script.

### FREVO with Gradle

An analysis showed that the current structure of FREVO, especially due to its component-plug-in-architecture, conflicts with the expected and possible project structure for Gradle.

### FREVO with Docker

Since FREVO and its components are open source under GPL V3, there was neither a legal nor a technical problem to put it into a virtual Docker container. We used an Ubuntu Linux system that was provided by Docker. Openjdk8 was installed as Java Runtime environment. After installing FREVO in the Docker system, it was pushed onto the free Docker Hub hosting platform (https://hub.docker.com/r/frodewin/frevo/, last visited: 2019-07-08). To reproduce a result made with FREVO it thus possible to (given that Docker is installed) download and execute the respective Docker container. In general, the result was easily usable, apart from some effort to get a graphical display working. The parallelization of simulation, which is a natural ability of FREVO, works fine as well inside a Docker container. The Docker image containing FREVO has a compressed size of 223 MB, which is mostly due to the files of Ubuntu Linux.

## LireSolr

LireSolr (*Lux & Macstravic, 2014*) is an extension for the popular Apache Solr (http://lucene.apache.org/solr/, last visited 2019-07-08) text retrieval server to add functionality for visual information retrieval. It adds support for indexing and searching images based on image features and is for instance in use by the World Intellectual Property Organisation, a UN agency, within the Global Brand DB (http://www.wipo.int/branddb/en/, last visited 2019-07-08) for retrieval of similar visual trademarks.

LireSolr brings the functionality of the LIRE library (*Lux & Marques, 2013*) to the popular search server. While LIRE is a library for visual information retrieval based on inverted indexes, it is research driven and intended to be integrated with local Java applications. Apache Solr is more popular than the underlying inverted index system, Lucene, as it allows to modularize retrieval functionality by providing a specific retrieval server with cloud functionality and multiple APIs to access it for practical use.

LireSolr is intended for people who need out of the box visual retrieval methods without the need for integrating a library in their applications. It can be called from any mobile,

server or desktop platform and runs on systems with a Java 8 runtime. This flexibility is valued among researchers as well as practitioners. LireSolr is hosted on Github (https://github.com/dermotte/liresolr, last visited 2019-07-08). Gradle and Docker build files are part of the repository.

### LireSolr with Gradle

The standard method of building LireSolr is by using Gradle. Current IDEs can import Gradle build files; any task can be done from within the IDE. While Gradle makes sure that the right version for each library is downloaded and everything is ready to build, installing the new features to the Solr server has to be done manually. The supporting task in Gradle just exports the necessary JAR files. The user or developer has to install Solr, then create a Solr core, change two configuration files, copy the JARs and restart the server to complete the installation. While these steps are extensively described in the documentation, it still presents a major effort for new users without prior experience of retrieval in general or using Apache Solr.

### LireSolr with Docker

As LireSolr is extending Solr by adding additional functionality, the intuitive way to create a Docker container is to extend the Solr Docker container. The *Dockerfile* defining the build of the Docker container is part of the LireSolr repository, where a specific Gradle task is building and preparing relevant files for the creation of the image. This includes the aforementioned JARs and config files as well as a pre prepared Solr core and a small web application as a client. The Docker container can easily be run and provides basic functionality for digital image search. Developers who just want to test LireSolr can get it running within seconds using Docker Hub (https://hub.docker.com/r/dermotte/liresolr/, last visited: 2019-07-08).

## ONE TOOL TO REPRODUCE THEM ALL?

In the previous sections, we presented tools for sharing software artifacts and examples showing how the tools can be applied in order to share scientific software artifacts. In this section, we now reflect on the advantages and shortcomings of the tools with respect to the results from our survey presented in the section "Survey".

Each of the presented tools has its pros and cons. For instance, the additional effort for sharing an artifact when using a build management tool is very low because in most cases a build management tool is used during the creation of the artifact. In contrast, it can be challenging and time-consuming for other researchers to get the build management tool up and running because required dependencies or the installation process may not be documented in detail. Software artifacts, which are provided as virtualized containers are easy to run and provide a high degree of security but are inconvenient in case a researcher wants to build upon previously published software artifacts.

When weighing these advantages and shortcomings we quickly see that *the one tool to reproduce all our scientific results* does not exist. Nevertheless, based on our findings from the survey we now want to give recommendations for creating reproducible results and scientific software artifacts which can be easily used by other researchers.

The survey clearly showed that many researchers are interested in building their research on the work of others, which becomes much easier, when published software artifacts can be reused. Furthermore, we saw that the average time researchers are willing to invest to get artifacts running is only about two days. Thus, we assume that it is very important for researchers to get the artifact running quickly, otherwise, researchers lose interest in using the artifact and start developing their own solution. When taking the demand for security into account, we see that virtualized containers appear to be a good choice. The provided software artifact can be executed without the overhead of installing it, by simply running the container. Furthermore, it is possible to become familiar with the artifact in the virtualized environment and check if the artifact is suitable to base own work on it.

Our findings mostly discuss the researchers' perspective working on original research questions. The role of a reviewer in a peer review process has not be discussed in the same detail, but we assume that it is similar to researcher's demands, but with even more demanding time constraints.

When researchers decide to build on the artifact, it may be cumbersome to continue using a virtualized container, because altering a software artifact is more convenient on a local system. This means that the researcher has to install the artifact locally, without virtualized container. According to our study, researchers currently prefer providing detailed instructions and build tools. Solely relying on this information, it could be challenging to install the artifact, as already discussed.

Dockerfiles are one solution to overcome this issue. As already explained, a Dockerfile is a kind of a construction guideline for Docker containers. It contains all command line directives, which are required to build a Docker container and can therefore be seen as exact procedure for the local installation of an artifact. Following the commands listed in the Dockerfile, local installation of a software artifact is relatively easy. These commands ensure that all dependencies are installed correctly, otherwise it would not be possible to create a Docker container. This means that by providing a Dockerfile, both options become possible, software artifacts can be executed in a secure container, but can also be easily installed by following instructions from the Dockerfile.

Another finding of our survey is that the long-term availability of software artifacts is important for researchers and should be around 10 years. In addition, the ACM Artifact Reviewing and Badging guideline (https://www.acm.org/publications/policies/artifact-review-badging, last visited 2019-07-08) emphasizes the importance of being able to reproduce results after a long time, by providing a separate badge for artifacts which are archived in archival repositories. When looking at our presented tools, we can see technical, as well as legal issues on the way to achieve long term availability. Although services, such as Code Ocean (https://codeocean.com/, last visited 2019-07-08) or Dryad (https://datadryad.org/, last visited 2019-07-08), are available for archiving software artifacts, the following points should be kept in mind. Tools such as Gradle rely on online repositories for downloading required dependencies. If only one of the required dependencies becomes unavailable, the build fails. This means that all dependencies, as well as all required tools have to be included when the artifact is archived. This leads to technical issues, because the

amount of required tools to reproduce a result could be tremendously high. For instance, if a required operating system or compiler is no longer available, the results can not be reproduced, which means that even these tools must to be archived. Besides this technical issue, packaging these tools could lead to legal issues as well when tools with limiting licenses are used. Furthermore, operators of platforms for archiving software could decide to discontinue service. This would result in loss of all artifacts archived by this provider.

## CONCLUSION

This article reflects on the reproducibility of research results in computer science. We collected the opinions and requirements of 125 researchers via an online survey. Focusing on our research question *RQ1* ("*To what extent is reproducibility of results based on software artifacts important in the field of computer science and in related research areas?*"), analysis of the survey's results confirmed our initial assumption that the reproducibility of research results is an important concern in computer science research. Besides, researchers not only want to reproduce results but also want to base their work on the results of others. The main reasons for the importance of reproducibility are improved credibility and improved understanding of results. Using established commercial models, as they are common for scientific publications, was seen as critical. Moreover, the majority of survey participants showed a willingness to use open source tools for making their results accessible and reproducible. Based on the researchers' opinions, we created guidelines which aid researchers in making their research reproducible. The applicability of various tools for publishing software artifacts was discussed while keeping our guidelines in mind. Scientific artifacts of different research areas in computer science were used to test the applicability of discussed tools for sharing reproducible research.

Regarding research question *RQ2* ("*What tools can be used to support reproducibility?*"), we identified a conflict between comprehensibility and simplicity of using a tool vs security measures avoiding to compromise one's system when testing foreign code. Available tools provide a variety of possible solutions, however, we could not identify a single tool fulfilling all requirements.

Finally, we discussed our findings and concerns on the process of publishing reproducible research. According to our study, the long-term availability of reproducible results is of great importance to many researchers, but we identified open issues in achieving availability for longer periods. Even if reproducibility of research is not common practice yet, we recognized a strong positive shift toward reproducible research, backed not only by individual researchers, but also by renowned scientific journals and publishers.

With this work already leading to new insights regarding reproducibility, it also installs a beachhead for future research. With the survey as input and the discussions regarding the interpretation we identified the context of a researcher as a hypothetically highly influential factor on the view on reproducibility. So how do for instance not only cultural, geographical, and project background of a researcher, but also the research area as well as the research communities influence the willingness to investigate extra time in making results reproducible? Future work could also address the question whether and to what extend project size would influence the willingness to invest time into reproducing work.

## ACKNOWLEDGEMENTS

We would like to thank all participants of the survey for their valuable input and all colleagues who helped us by sharing their practical experience and discussion. We thank the anonymous reviewers for their constructive comments on a previous version of the article. We are grateful to Lizyy Dawes for proofreading an earlier version of this article.

### Funding

This work was supported by the Austrian Science Fund (FWF) under the CHIST-ERA project 496 CONCERT (project no. I1402). There was no additional external funding received for this study. The funders had no role in study design, data collection and analysis, decision to publish, or preparation of the manuscript.

### Grant Disclosures

The following grant information was disclosed by the authors:
Austrian Science Fund (FWF) under the CHIST-ERA project 496 CONCERT: I1402.

### Competing Interests

The authors declare that they have no competing interests.

### Author Contributions

- Wilfried Elmenreich conceived and designed the experiments, performed the experiments, analyzed the data, contributed reagents/materials/analysis tools, performed the computation work, authored or reviewed drafts of the paper, approved the final draft.
- Philipp Moll conceived and designed the experiments, performed the experiments, analyzed the data, contributed reagents/materials/analysis tools, prepared figures and/or tables, performed the computation work, authored or reviewed drafts of the paper, approved the final draft.
- Sebastian Theuermann conceived and designed the experiments, performed the experiments, analyzed the data, contributed reagents/materials/analysis tools, performed the computation work, authored or reviewed drafts of the paper, approved the final draft.
- Mathias Lux conceived and designed the experiments, performed the experiments, analyzed the data, contributed reagents/materials/analysis tools, prepared figures and/or tables, performed the computation work, authored or reviewed drafts of the paper, approved the final draft.

### Data Availability

Links to the tools used are available in the article. This includes data and code produced by us as well as code from others where we have built upon.

The following repositories contain parts deposited for the article:

Docker Images:

https://hub.docker.com/r/phmoll/saf-prebuild/

https://hub.docker.com/r/dermotte/liresolr/

https://hub.docker.com/r/frodewin/frevo/

We used a version of FREVO software, Sourceforge:

http://frevo.sourceforge.net/.

## Supplemental Information

Supplemental information for this article can be found online at http://dx.doi.org/10.7717/peerj-cs.240#supplemental-information.

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
