# Peer review of "Making simulation results reproducible—Survey, guidelines, and examples based on Gradle and Docker"

_PeerJ Computer Science, doi:10.7717/peerj-cs.240_

## Round 0.1 · original submission · Major Revisions

After a new review of the revised manuscript with a fresh set of reviewers I am confident that the paper is now in good shape for PeerJ CS, assuming that the reviewer's major comments are addressed. Particularly, I urge the authors to further tidy up their description of the methodology and experimental design.

However, please carefully address all reviewer comments or provide arguments in a response letter why they should not be addressed (note that it's ok to disagree with the reviewers, as long as good arguments are brought forward).

Reviewer 1 ·

Basic reporting

The paper is well-written. The general structure is fine as well, only within certain sections I would prefer more structure. All the survey results are discussed in Section 2.3, which is a massive wall of text. I would recommend adding some subsections, for example, a separate section for the results related to peer reviewing.

Intro and related work are ok. I would add a brief summary about the main findings at the end of the introduction.

Figures are relevant and also well described, but in my opinion, the figures are way too small. When printed the labels are hard or near impossible to read. Make sure to use proper font sizes.

Raw data was supplied and seems proper.

Experimental design

The research conducted is in scope of the journal, the research questions are relevant and meaningful. The survey part including the methodology definitely improved compared to the previous version. However, there is still missing information:
-) How did the authors come up with both the survey questions and the possible responses for single-choice and multi-choice questions? Predefined answers increase the chance for introducing researcher bias, which is also not reported as a potential threat in the threats to validity section. The added threats to validity section needs to be more thorough. I would recommend the authors to check out related literature. The already mentioned Runeson et al. “Case study research in software engineering: guidelines and examples” is a good starting point.
-) There is no information how open / free-form questions were analyzed. Was this based on open coding?
-) There is no information whether all questions were mandatory for survey participants. We do not learn how many participants completed the survey, for example, did all 125 participants complete the survey, or are there some partial results as well?
-) We do not learn how the authors treated numerical inputs. Were some ranges (e.g., in years) provided upfront? Or were these ranges and categories introduced later on?

Learning about subpopulations is helpful, I really appreciate that we now have this information. However, I’m wondering how ‘relevant’ those 11 bachelor and master students are for a survey on the reproducibility of research results, especially for those parts covering reproducibility in the context of peer review.

Validity of the findings

What I’m missing throughout the entire paper (it is only mentioned on the last page) is a discussion about the newly introduced artifact evaluations as part of some computer science conferences/journals. The goal is to make research results reproducible and badges awarded for papers also provide some sort of credibility / recognition. So, the question is whether these new processes are enough, or do we still need to think about ways for including artifact evaluation during peer review. I would have appreciated to have some discussion when reflecting on the survey results and proposing the general guidelines in Section 3.

Regarding the third guideline in Section 3, even though source code is provided and not just binaries, the code can still contain malicious parts.

“The average amount of time participants would invest in making software of others to reproduce results was 15.94 hours, neglecting 5 outliers who would spend 100 or more hours.” I’m a bit confused why those 5 outliers are excluded from the average?
Especially, since the average time spent for making the own research results reproducible is 24.4 hours and 6 participants that invest between 100-250 hours seem to be included? Without having the exact numbers, it seems that both averages could be on a very similar level? I recommend the authors to clarify why they exclude / include certain data points.

I appreciate that the authors summarized the comparison of their examples in a table. However, I have the impression that categories and ratings are very subjective. Compared to the other approaches, why is the effort for sharing Docker images ‘moderate’? Once you have the image (and we are only considering the sharing effort) it is just a single command to upload it to Dockerhub. How is this more effort than pushing for example a Node.js package to NPM?
Furthermore, based on the description of the categories it is hard to tell the difference between “Required software” and “Effort for installation and execution”. I recommend the authors to go over their categories again, make them clearer and not overlapping, and then do the ratings/assessments again and try to justify ratings/assessments.

Additional comments

The authors conducted a study among researchers with the goal to identify the importance of reproducibility of research results and current issues for making software artifacts available and reusing software artifacts provided by others. Further, the authors provided a set of guidelines for reproducibility and discussed and compared existing tools for sharing and reproducing software artifacts.

Even though the paper improved compared to the previous version, there are still many major flaws detailed in the comments that need to be addressed before it can be considered to be published in this journal.

Further comments:
Related work improved, but there is no statement on how this paper goes beyond the findings reported by others. What is the research gap the authors are trying to fill?

The authors need to be careful with the Docker terminology. There is a difference between Docker images and Docker containers. You do not share containers, you share images. Images become containers when they run on the Docker engine. Furthermore, also when using Dockerfiles to specify the ingredients and the recipe on how to build an image, the source code and all the other artifacts that are needed for building an image need to be available, either as part of the host image or (finally) on the machine where the image is built. Consequently, when “sharing” artifacts with Dockerfiles, it is (in most cases) not just the Dockerfile that needs to be shared.

Gradle is not necessarily bound to specific project structures. This is only the case when using the default settings, but Gradle is highly customizable and thus the configuration can be adapted to support building projects with different structures. The authors should consider that when assessing Gradle.

Section 2.3: “These responses indicate the need to provide open source methods of sharing software artifacts free of charge”. Maybe this goes down to some lack of rigor in the survey design, but there are various platforms out there that allow sharing artifacts completely free of charge and also supporting large file sizes. For example, Zenodo (https://zenodo.org/) and figshare (https://figshare.com/).

Minor:
First sentence sounds a bit clumsy. Maybe “The paper addresses two [research] questions related to reproducibility within the context computer science research”?
Line 33, “paper addresses two questions related to”, maybe “…two research questions…”?
Line 34, “… with the the aspects…”

·

Basic reporting

- The language is ok, and the article flows without any specific issues. The correct use of the Saxon genitive should be revised (e.g., Docker's vs. Docker concepts)

- The references to the literature are sufficient. The authors identify the work of Cito et al. as very closely related to theirs. Accordingly, I suggest them to better pinpoint the differences between the two.

- Please, check the reference style. Avoid brackets in (Author (Year)) format.

- I was not able to find a link to the raw data from the survey.

- The introduction is quite weak and requires more context before introducing the research questions.

- In the Introduction the authors state "Reproducing results of published experiments, however, is often a cumbersome and unrewarding task. The reason for this is twofold" But then only one reason is reported, "First, some fields ..."

Experimental design

The survey design is detailed enough but suffers from a few shortcomings.

RQs
- The first research question implies that for the sample reproducibility may not be important. - Is this a good sample for surveying best practices and tools (i.e., second RQ)?
- Please, specify what are the "related research areas of computer science."

Sample
- Please, report (even a ballpark) of the total amount of people you may have reached in your survey (not only the ones who answered).

- The characterization of the respondents does not take into account overlaps (e.g., one can be a Ph.D. student and work as a researcher within a project).
- What is the area of research of the remaining 13% of the sample not reported in Section 2.2?
- It would be beneficial to provide a geographical characterization to check whether cultural factors could impact the findings.

- The authors later report that 35 out of 125 respondents never published a software artifact. This information should be reported earlier. Most importantly, this should be elaborated in reference to my previous comment regarding the validity of the sample.

Results
- Notice that time one is willing to invest in reproducibility of a project depends on the size of the said project. Accordingly, a characterization of the projects the respondents work on would be beneficial to interpret this result.

- The language is "leading." Avoid adjectives as "interestingly" or "only 28 out of 125." Let the reader decide what is interesting and whether 28 is "only" a small proportion of 125 (for me, it is not).

- No statistical analysis was performed. Hence, the deducing facts about the world based on these numbers only is quite subjective. I recommend performing statistical analysis (at least correlations) to support some of the claims. Moreover, avoid statements like "There was no significant difference determined between different research environments." as no significance test was performed.

Validity of the findings

- Data from the survey is not provided (or at least I could not find it) and no statistical test is performed.

- I am not sure how the guidelines and tools follow from the results of the survey. The former are quite general (which is fine), the latter should be better contextualized in specific usage scenarios (and if possible, discipline related to CS)

- Please explain what is a Dockerfile.

Additional comments

- Consider moving inline links to footnotes for better readability.

---

## Round 0.2 · Minor Revisions

As you can see from the comments, the scientific reviewers are now largely happy with the manuscript. However, there are still some recommendations for presentation-level improvements. Please incorporate what you think makes sense from these suggestions.

Please also note that one of the reviewers has uploaded an annotated PDF rather than providing their comments in text.

Reviewer 1 ·

Basic reporting

The paper is well-written. Introduction and related work definitely improved compared to the previous version. The motivation of the study and how the conducted research differs from related work is now much clearer.

Figures are relevant and also well described, but in my opinion, some figures are still too small (e.g., Fig 1, 3, 6, 7). The labels are easier to read compared to the previous version, but there is some space for improvement. Consider an increased font size.

The structure mainly stayed the same. I still have the impression that Section 3.3 (formerly 2.3) requires some sub-sections or paragraphs emphasizing the covered topics. I’m also not a fan of the title “What Researchers Want”. Why not having some smaller paragraphs such as relevance, effort, peer review, publication & security, or similar? This would break this wall of text and provide more structure.

Raw data was supplied and seems proper.

Experimental design

The research conducted is in scope of the journal, the research questions are relevant and meaningful. The entire experimental design and the methodology of the survey part improved and threats to validity were updated accordingly. The authors addressed most of the comments sufficiently.

I’m not overly convinced on the relevance of bachelor or master students on a survey regarding the reproducibility of research results. The argument provided in the author’s response letter “we expect the opinion and knowledge of the subject of today’s students being relevant for the next generation of researchers” does not fully convince me, especially considering the fact that only a small percentage of students conduct PhD studies afterwards. For me it would be ok to include these students as long as there is somehow an indication that they are or were involved in research projects. However, answering questions on reproducibility and how much time and effort you would invest in making your own results reproducible and (re-)using the tools of others without having first-hand research experience seems problematic.

Validity of the findings

The authors clarified and addressed most of the raised concerns appropriately.

I’m only missing information on how the authors decided on cluster sizes (e.g., page 5: “Summarizing the results in clusters…” and “Summarizing through clusters…”) and why do for example the cluster sizes on the time willing to invest for reproducing results of others and on the time for making own results reproducible differ?

Additional comments

The authors conducted a study among researchers with the goal to identify the importance of reproducibility of research results and current issues for making software artifacts available and reusing software artifacts provided by others. Further, the authors provided a set of guidelines for reproducibility and discussed and compared existing tools for sharing and reproducing software artifacts.

The paper definitely improved, most of the issues are addressed. As described, there are some minor comments I would like to see addressed before publishing the paper in the journal.

Further comments:
For the previous revision I pointed the authors to the correct usage of Docker terminology. Most of this image vs. container mix-ups have been fixed, but there are some more occurrences, e.g., page 10 lines 364 and 373, page 13 lines 517 and 519, page 15 line 607. I recommend the authors to conduct thorough proof-reading to fix these as well.

I don’t see much value in the added correlation analysis. I have the impression that it is pure number reporting and it is not really used to support some of the author’s claims such as the mentioned “We could not identify significant differences between different research environments”. I would just recommend the authors to be careful with the language and avoid the usage of terms such as “significant” as long as there is no underlying statistical test.

Minor:
Page 7, line 262: “A multiple-choice question…”, this sentence seems incomplete.
Page 7: I would recommend merging the pre-defined answers (starting at line 265) with the frequencies they were selected (line 272). Further, I would remove the listing of the additional five “other” answers in detail and just provide a categorization (e.g., dependencies).

·

Basic reporting

The language was improved and, overall, the text flows better.
See my annotated PDF for language and comprehensibility improvements.

The introduction section has been improved considerably, which improves further understandability and scoping of the paper.

Experimental design

My previous request to perform statistical analysis was carried out to a good extent and with sufficient information to fairly evaluate the validity of the results.

Validity of the findings

Data is provided. The findings are used to answer the research question (although not formally, see PDF annotations).

Additional comments

I would like to thank the authors for their work.
Please, see my comments in the PDF for further, minor, improvements.

---

## Round 0.3 · accepted · Accept

It is my opinion that the paper is now ready for publication. I congratulate you on this excellent research!